# Effect of Graphene Carbon Nitride on Ultraviolet-Curing Coatings

**DOI:** 10.3390/ma13010153

**Published:** 2019-12-31

**Authors:** Zusheng Hang, Huili Yu, Yan Lu, Xu Huai, Lingpeng Luo

**Affiliations:** 1Jiangsu Key Laboratory of Advanced Structural Materials and Application Technology, Nanjing 211167, China; whisun@126.com; 2School of Material Engineering, Nanjing Institute of Technology, Nanjing 211167, China; y1826432343y@163.com; 3Institute of Advanced Synthesis, School of Chemistry and Molecular Engineering, Nanjing Tech University, Nanjing 211816, China; 662081704021@njtech.edu.cn; 4International Academy of Optoeletronics at Zhaoqing, South China Normal University, Guangzhou 526238, China; lingpeng.luo@zq-scnu.org

**Keywords:** UV-curable coatings, g-C_3_N_4_, composite nano-coatings, oxygen inhibition

## Abstract

Graphene carbon nitride (g-C3N4) was successfully prepared by semi-closed pyrolysis and then incorporated into the ultraviolet (UV)-curing system to synthesize different specimens of g-C_3_N_4_-hybridized UV-curing (g-C_3_N_4_/UV) coatings. The apparent appearance and dispersity g-C_3_N_4_ were characterized through X-ray diffraction (XRD), scanning electron microscopy (SEM), and Fourier transform infrared spectroscopy (FTIR). The influence of the curing speed and mechanical properties was also tested. The dispersion level of g-C_3_N_4_ can be kept less than 1 μm by mechanical mixing. The pencil hardness of composite coatings could be 6H while the adhesion based on glass could be 1 level. The degree of curing of the coating could be obviously improved by the addition of g-C_3_N_4_, leading to a 7 percent increase of the gel content. Additionally, the decomposition of hard segments of polyurethane acrylate could be avoided by the use of g-C_3_N_4_ resulting in an increasing stability to heat. We found the gel content in an aerobic environment was lower than that in an anaerobic environment. On this basis, the function and mechanism of g-C_3_N_4_ was investigated in detail and methods to eliminate the O_2_ were proposed.

## 1. Introduction

UV curing technique is becoming more and more important in coating due to its various advantageous characteristics: high machining speed, lower energy consumption and costs, chemical inertness, and environmental friendliness [1,2]. 

For application, it is quite necessary that photoinitiators, light stabilizers, and other light activated chemicals to coexist in the same reaction system [3,4]. However, this presents a contradiction as polymerization of photo initiation agents may be initiated by the absorption of ultraviolet radiation, while the light stabilizers simultaneously absorb ultraviolet light to prevent deterioration of the film. Such a competition effect of UV absorption may lead to many poor mechanical properties, such as adhesion, hardness, and thermal stability, rendering it less useful for application. In addition, even in the presence of optical activator, it is difficult for traditional photoinitiators to achieve effective control of the curing rate in the UV curable process [5,6]. Therefore, considering these aforementioned two problems, developing new types of photo initiator and light stabilizer to coordinate the photostabilization of UV-cured coatings has drawn considerable attention in recent years.

An effective method for increasing the photocatalytic efficiency and mechanical properties is to add nanoparticles (1–100 nm), to prepare hybrid UV-curable coatings [7,8]. It shows considerable improvement and novel performance compared with traditional composites [9,10]. Liu et al. [11] added the obtained functional (TiO2-SiO2/P(MMA-co-PMPM)) fillers to polyurethane acrylate (PUA) oligomers to get UV-curable nanocomposite coatings. Experiments show that composite nanoparticles reduced the UV-curable PUA coatings degradation rate dramatically, thereby improving the UV aging resistance of coatings. Recent studies found that a new type of carbon material—graphitic carbon nitride (g-C_3_N_4_)—shows certain favorable properties when applied to UV curing [12]. g-C_3_N_4_ is a layered material similar to graphene and composed of only C, N, and some impurity H. It is a two-dimensional conjugated polymer with abundant s-triazine (C_3_N_4_) functional groups on the surface. Such a structure could form a π-conjugated plane such as that of graphite, which could reasonably induce a charge transfer and produce a synergistic effect of curing with initiators in the UV irradiation [13]. Meanwhile, g-C_3_N_4_ has other attractive advantages such as stable physicochemical and unique electrochemical properties as well as high quantum yield, thermal stability, and organic compatibility [14,15,16,17] Thus, it can play a considerable role in stabilizing matrix structure when applied in various fields [18,19,20]. Li, Yeping et al. [21] synthesized a g-C_3_N_4_ modified Bi_2_O_3_ (g-C_3_N_4_/Bi_2_O_3_) composite by a mixed calcination method. The g-C_3_N_4_/Bi_2_O_3_ composites show high visible-light photocatalytic performance. The enhanced photocatalytic activity is mainly attributed to the effective separation of electrons and holes by the addition of g-C_3_N_4_. Accordingly, the contradiction of light initiation and chain termination in the UV curing process can be effectively solved by the development of new high performance UV-curable coatings incorporated with g-C_3_N_4_, which can play the dual role of photo initiators and light stabilizers, thereby improving the film aging resistance with a high polymerization rate.

In this work, the g-C_3_N_4_ was synthesized by semi-closed pyrolysis then compounded with other known constituents of UV-curable coatings to synthesize g-C_3_N_4_-hybridized UV-curing (g-C_3_N_4_/UV) compound coatings. The physical and chemical properties of g-C_3_N_4_/UV compound coatings were characterized by X-ray diffraction (XRD), scanning electron microscopy (SEM), Fourier transform infrared spectroscopy (FTIR), thermo gravimetric (TG), and differential thermal gravity (DTG) methods. The effect of g-C_3_N_4_ ratio on the hardness, gel content, acid and alkali resistance, adhesion, and thermal stability properties of hybrid coatings were also discussed in detail. The influence of temperature, substrate, and irradiation distance on the properties of g-C_3_N_4_/UV compound coatings was also explored. The effect of oxygen on the curing properties of the composite coatings was analyzed and methods for avoiding oxygen inhibition were proposed. Lastly, a possible mechanism of g-C_3_N_4_ accelerating an otherwise kinetically hindered reaction associated with g-C_3_N_4_/UV composite nanocoatings was discussed.

## 2. Experimental Details

### 2.1. Preparation of g-C_3_N_4_

g-C_3_N_4_ was firstly prepared by heating a mass of melamine in a 50 mL muffle furnace at 550 °C for 2 h with a ramp rate of 10 °C min^−1^, and then cooling to 50 °C at the same rate for another 2 h followed by a sequential decrease to room temperature. The obtained pale yellow lumps were placed in an agate mortar and ground until no grainy feeling. The products were soaked with ethanol and ultrasonic dispersion for 30 min. After drying, the products were washed with deionized water for 3 times and dried under vacuum at 70 °C for 6 h, and ground into yellowish powder.

### 2.2. Preparation of Compound UV-Curable Coatings

To prepare the hybrid coatings, the as-synthesized g-C_3_N_4_ powders were directly mixed with 10 g of UV-curable coating and dispersed by magnetic stirring at 6000 r/min for 60 min and ultrasonic dispersion for 1 h at room temperature. It should be noted that 1/3 proportion of water in digital homoeothermic water bath should be exchanged to avoid undesirable environmental factors. Finally, the mixture was stirred with a high-speed shearing machine and the coatings were obtained.

### 2.3. Characterization

The film samples were uniformly coated by wire wound rods on thin microscope slides under a high-pressure mercury lamp during the appropriate UV dose. 

#### 2.3.1. XRD

The XRD spectra was recorded using a Rigaku D/MAX2500 PC diffractometer with Cu K radiation, with an operating voltage of 40 kV and an operating current of 100 mA.

#### 2.3.2. FTIR Spectra

The FTIR spectra of samples before and after UV curing were scanned using a Magna-IR 550 spectrometer (Nicolet Instruments, Madison, WI, USA). In this study, sandwich-like NaCl plates were used for FTIR scanning to minimize the influence of oxygen in the peripheral atmosphere. 

#### 2.3.3. SEM Observation

SEM micrographs were captured with a Hitachi H-600 apparatus (Hitachi Corp., Tokyo, Japan). Samples of composite coatings were prepared by casting the solution on substrates and coated using aurum spattering. 

#### 2.3.4. Thermogravimetric (TG) Analysis 

TG curves were obtained using a thermogravimetric apparatus (SDT 2960, TA instrument, New Castle, DE, USA). The temperature ranged from the 23–850 °C with a heating rate of 10 °C/min in a dynamic nitrogen flow of 50 mL/min. 

#### 2.3.5. Pencil Hardness

The pencil hardness test was performed using a Sheen pencil hardness kit based on ASTM D3363. The grades reported are the hardest pencil grades that failed to scratch the surface of the coatings. 

#### 2.3.6. Adhesion

Film adhesion of the coating was measured using the crosshatch adhesion tape test method, according to ASTM D3359. A crosshatch consisting of 11 intersecting vertical and horizontal lines was carved on the coating using a blade. The lines were carved 1 mm apart, and a brush was used to gently remove any detached flakes or ribbons from the coating. A self-adhesive tape was then placed on the crosshatch and gently rubbed to allow optimum contact between the tape and coating. The tape was then peeled off in a single stroke at an angle close to 180 degrees, and the grid area was inspected for any film removal. The methods for the film adhesion tests were adopted from reference [22]. 

#### 2.3.7. Acid and Alkaline Resistance

Acid and alkaline resistance property was tested by soaking the 0.5 mm thick films in 5% w/w HCl solution or 5% w/w NaOH aqueous solution, respectively. The film condition of the coatings after every 2 h of immersion was contrasted with the ones before and recorded. 

#### 2.3.8. Gel Content 

Gel content of the cured coatings was measured by immersing a known weight of 0.5 mm thick films, peeled off from the glass plate, m_1_, in xylene for 24 h. The coatings were removed and dried in an oven at 70 °C for 24 h before measuring the dried weight, m_2_. The gel content of the cured coatings was determined by the following equation:(1)$$\omega =\frac{m_{2}}{m_{1}}\;\times \;100\%,$$
where m_2_ is the weight after extraction and m_1_ is the weight before extraction. 

#### 2.3.9. Tack Free Time 

The tack free time was physically examined by using the method of thumb or finger tack testing to determine the condition of the films under UV irradiation. The UV light used for curing was from a 400 W lamp emitting 225 mW/cm^2^ of UV radiation (k = 365 nm). 

## 3. Results and Discussion

### 3.1. g-C_3_N_4_

#### 3.1.1. XRD

Figure 1 shows the XRD pattern of pure g-C_3_N_4_ obtained by thermal decomposition. The pure g-C_3_N_4_ exhibited two obvious positions of diffraction peaks at 13.1° and 27.4°, which can be respectively indexed as the (100) and (002) crystal plane. <111> plane corresponds to a nitric distance among triazine of d = 0.685 nm, and <002> plane indicates a laminar stack with a distance of d = 0.32 nm. The XRD patterns are consistent with the reference suggested that the intrinsic crystal structure of pure g-C_3_N_4_ had been maintained.

#### 3.1.2. FTIR

Figure 2 shows the FTIR spectrum of pure g-C_3_N_4_ prepared by semi-closed pyrolysis. The typical absorption peaks of triazine units at 802 cm^−1^ are observed. The intense band at 1200–1800 cm^−1^ corresponds to aromatic CN heterocycles. For the compound sample, the absorption peaks at 1630, 1420, 1330, and 1250 cm^−1^ could be attributed to the stretching vibrations of C=C, the scissor vibration of =C, and the stretching vibrations of the conjugated π units of C–N and C=N, respectively. The broad band at 3175–3460 cm^−1^ is characteristic of the N-H stretches, which could be explained by incomplete condensation. All above results illustrate that the products possess a thiotriazinone structures.

### 3.2. g-C_3_N/UV Compound Coatings

#### 3.2.1. Dispersivity

SEM micrograph of different content of g-C_3_N_4_ in compound coatings was shown in Figure 3. Among them, Figure 3a showed that 0.5% of g-C_3_N_4_ had the best dispersion. The average particle size could be 0.5–2 μm. However, as shown in Figure 3b–d, with the increment of the g-C_3_N_4_, the particle size increasingly reached 10 μm or even more than 50 μm. The results indicated that the agglomeration was severe with a bad dispersion when the additives exceeded 5%. According to the result, the best dispersivity of g-C_3_N_4_ in the compound coatings occurred at 0.5%, therefore we synthesized an organic compound coating with a proportion below 0.5% (the upper limit of the range) of g-C_3_N_4_ for the further research.

#### 3.2.2. Composition

Figure 4 shows the FT-IR spectrum of pure UV-cured coatings (b) and compound coating with 0.3% of g-C_3_N. As shown in Figure 4a, the absorption peak at 2350 cm^−1^ corresponded to the typical conversion of –NCO– bonds in UV-curable coatings, which proved that the polymerization of UV-cured coatings could be nearly 100% under visible light. The intense absorption peaks at 1160 cm^−1^ represents the stretching vibration of C–O–C bonds, thereby demonstrating that the polyurethane polymers were converted completely. The representative peaks at 1631 cm^−1^ and 810 cm^−1^ representing the stretching vibration of C=C bonds and the out of plane of C–H bending mode of acrylate were still observed. The results imply that the photopolymerization of pristine UV coatings could not be sheer under ultraviolet light. The main reason was that the sustained polymerization increased the viscosity of the system, leading to an increasing resistance to chain segment movement beyond the impetus of self-acceleration, which would result in a low degree of polymerization and incomplete reaction.

In comparison of Figure 4a,b, the absorption peak at 2350 cm^−1^ that corresponded to the conversion of –NCO– bonds essentially disappeared, while the typical peaks that detect the presence of acrylate at 1631 cm^−1^ and 810 cm^−1^ became reduced in intensity. These certified a beneficial improvement in the degree of photopolymerization after g-C_3_N_4_ was added.

The intense band at 3180–3420 cm^−1^ that characterized the conversion of the amino group. The absorption peaks at 1680 cm^−1^ and 1406 cm^−1^ corresponded to the stretching vibration of CN heterocyclic bonds. The typical absorption peaks of the breathing mode of the triazine units at 806 cm^−1^ showed in Figure 4b jointly attested to a perfect amalgamation between g-C_3_N_4_ and the other UV systems, indicating the successful synthesis of compound UV coatings.

#### 3.2.3. Hardness and Abrasion Resistance

To measure the hardness of the UV-curable coatings, coatings with a wet thickness of 100 ± 5 μm were applied and the Shore D hardness was determined according to the testing method of ASTM D3363. The measured results for the hardness of composite coatings with stepwise changing ratios were showed in Figure 5. The apparent hardness of the coating increased at first, and then decreased, which reaching a maximum at 6H when 0.4% of the g-C_3_N_4_ was supplied. Notably, there was a positive correlation between the abrasion resistance and hardness of UV-curable coatings. 

Accordingly, more uniform dispersion of g-C_3_N_4_ means the greater the pencil hardness of the cured films, as well as the higher the abrasion resistance. This is because g-C_3_N_4_ had good compatibility with the organic urethane acrylate and could form interpenetrating networks between the three-dimensional layered structure. The crosslinking density of UV-curing systems increased due to physical entanglements similar to that obtained by a small amount of g-C_3_N_4_ aggregation, which produced a strong interaction and hindered the movement of molecular chains, and consequently reinforced the hardness and abrasion resistance. Considerably, a large amount of g-C_3_N_4_ significantly increased agglomeration and deteriorated organic compatibility, with a sharp decline in interfacial interaction, hardness, and wear resistance due to stress concentration. This is evident from Figure 3c,d.

#### 3.2.4. Adhesion

We examined film adhesion with glass as a curing base. Transparent glass helps avoid experimental error caused by different depth of scratches, while the smooth surface can rule out the influence of physical attachment caused by the uneven substrate surface.

Figure 6 shows the variation of adhesion between the glass and the completely cured composite coatings with different composition. With the increasing of g-C_3_N_4_ content, the adhesion between the coating and the glass increased first and then decreased. Correspondingly, the adhesion was good initially while it became worse then. When the g-C_3_N_4_ content reached approximately 0.3%–0.4%, the adhesion could reach level 1, with 1% being equivalent to the level of pure film. Since there was no polymerization reaction during the curing process that involved forming chemical bonds with the substrate, therefore, the film adhesion resulted from the formation of hydrogen bonds between the hydroxyl groups of the substrate and a small amount of water vapor, and also depended on the shrinkage stress generated by volume shrinkage during the curing process. When dispersed well in the matrix, the large quantity of basic groups in g-C_3_N_4_ increased the number of polar groups in the curing system, thereby increasing the interaction with the substrate. Simultaneously, the fine particulate g-C_3_N_4_ may hinder the movement of polymer chains to significantly reduce the volume shrinkage as well as shrinkage stress. This was beneficial to the improvement of adhesion. When g-C_3_N_4_ is added in excess, it will partially settle at the bottom of the system and agglomerate to affect the adhesion properties.

#### 3.2.5. Acid and Alkaline Resistance

According to the hardness and adhesion testing, sample with 0.3% of g-C_3_N_4_ was chosen to characterize acid and alkali resistance. The coated glass was immersed into the 10% HCl and 10% NaOH solution, respectively. The results were shown in Table 1 and Table 2. The existence of g-C_3_N_4_ is beneficial in enhancing the alkali resistance of composite coatings, but reduces the acid resistance. The surface of g-C_3_N_4_ contains a large number of basic groups, it is easy to interact with the acid and then protonize into salt, so that its solvent resistance decreases and the curing system is easily separated from the system. As the immersion time progresses, acid resistance declines with a decrease in the crosslink density. On the contrary, in an alkaline environment, g-C_3_N_4_ could improve the alkali-resistance, thus having no polymerization interaction with the interface and enhancing the crosslinking density.

#### 3.2.6. Gel Content

Figure 7 shows the gel content for different proportions of g-C_3_N_4_. We could see that the gel content firstly increased and then decreased with g-C_3_N_4_ addition, reaching the maximum value of 83% when the g-C_3_N_4_ content was approximately 0.2%. The main reason is that when the proportion is less, its efficient dispersion in the polymer provided strong interaction during the polymerization, which limited the movement of the macromolecular chain and enhanced the crosslinking density of the coating. When the proportion of g-C_3_N_4_ is increased, a sharp decline in particle dispersion and obvious agglomeration reduced the interaction and crosslinking density, which caused it to be susceptible for swelling for that the organic solvent molecules easily enter the macromolecular chains, resulting in a decrease of the gel fraction of the cured film.

#### 3.2.7. Thermal Stability

The quantitative analysis of the effect of g-C_3_N_4_ on the thermal stability was conducted by TGA, which was performed from room temperature to 600 °C by heating at 20 °C/min. The TG curves of some g-C_3_N_4_/polyurethane acrylate (PUA) compound coatings containing different proportions of g-C_3_N_4_ were illustrated in Figure 8. Among them, Figure 8a–d corresponds to samples with 0%, 0.1%, 0.2%, and 0.4% of g-C_3_N_4_, respectively. From the differential curves of TG, the temperature at which there is a maximum mass loss rate (T_max_), as well as the decomposition onset temperature T_onset_ is considered a parameter for the estimation of the thermal stability influenced by the proportion of g-C_3_N_4_. For the four curves showed in Figure 8a–d, there are two weight loss stages and each one relates to the basic structure of polyurethane acrylate.

Polyurethane acrylate is synthesized by the condensation polymerization of polyester or polyether polyol and diisocyanate, during which the soft segments were constituted by COC backbone of polyols, while the hard segments were composed by the crosslinking hydrogen bonds from urea, ureido, and carbamates generated by the reaction of water and other small molecules. Consequently, many excellent properties are obtained the alternate arrangement of the soft and hard segments.

The first weight loss can mainly be attributed to the breakage of hydrogen bonds among the hard segments, where ammonia, nitrogen, water vapor, or carbon dioxide were released resulting from the thermal decomposition of certain groups. The sharp weight loss can probably be attributed to the fracture of macro-molecule chains and the breakage of crosslinking structures. A further weight loss occurred when the polymer is decomposed into several small molecules, leaving the final residues of the char.

It can be seen from the TG diagram of Figure 8 that the weight loss rate of the first stage of the substrate was reduced from 20% of the 0% component to 14.4% of the 0.4% component. When the weight loss rate was 10%, the degradation temperature of pure paint and 0.4% g-C_3_N_4_ composite coating was 175 °C and 221 °C. Thus, the g-C_3_N_4_ is exceptionally useful in improving the thermal stability. This results from the amino interaction of g-C_3_N_4_ with partial hard segments groups containing nitrogen and hydrogen, increasing the density of hydrogen bonds, and the degree of crosslinking. On the contrary, in the second weight loss stage, the compound coatings containing g-C_3_N_4_ exhibits a mass loss rate higher than that with no additives with a common concluding temperature of approximately 472 °C, which contributes to the irrelevancy between g-C_3_N_4_ and C–O–C bonds in polyester or polyether polyol. Additionally, the uniformly dispersed particles in the system increase the distance between the polymer chains. More evidently, in the pyrolysis of hard segments, the original entangled backbone becomes loose, and the curing system disintegrated easily during the process. All above contributed to a faster weight loss rate.

### 3.3. Mechanism

It is clear that the effect of O_2_ inhibition on the urethane acrylate curing system results from the air exposure during UV curing. Here, the performance of several cured coatings were tested under aerobic and anaerobic conditions, and the mechanism of oxygen inhibition on polymerization was put forward. In addition, several protective measures were concluded to protect the cured films from the O_2_ inhibition effect.

#### 3.3.1. Influence of Oxygen (O_2_) Inhibition on the Properties of Cured System

##### ① Mechanical Properties

The pencil hardness and adhesion of fully cured coatings with the same ratio were tested under aerobic and anaerobic conditions respectively (Table 3). The pencil hardness of coatings, either pure or composite ones, complete curing under the aerobic environment was lower than that under anaerobic, as evidenced by the drop from 3 to 6 H and 2 to 5 H, indicating that the mechanical properties of the coating would be severely affected by the oxygen.

The competitive consumption of free radicals by O_2_ could affect the further polymerization of monomers, leading to the termination and reduction in degree without curing to form crosslinked structures, leaving a part of the unreacted monomer resulting in a similar adhesive viscous state. Besides, the potentially dissolved O_2_ in the coatings would further affect the deep curing, making the concrete soft and difficult to dry even though the surface is non-stick. The change in adhesion under aerobic and anaerobic conditions was not obvious, contributing to the closer contact of the coatings and substrates, which made it difficult for O_2_ immersion to capture free radicals, promising a substantially unaffected bottom of the coating.

A slight decrease in the adhesion of the composite coatings might be caused by two reasons. One is the partial aggregation of g-C_3_N_4_ produced a gap between the bottom coating and the substrate, resulting in the penetration of some O_2_ into the matrix. The other is a small part of g-C_3_N_4_, with size in the nanometer range, produced photo-generated electrons and holes and then captured O_2_ to generate oxygen-free radicals to cause substrate oxidation. All of this eventually caused some PUA aging in the curing system and the substrate bonding effect is poor with reduced adhesion.

##### ② The Tack Free Time

Tack free time, a parameter to characterize the curing level of coating surface, can be used to estimate the influence of oxygen inhibition effect while extending the tack free time meant higher oxygen inhibition.

The tack free time of pure and composite coatings in aerobic and anaerobic conditions respectively are shown in Table 4. It was very interesting to see that the tack free time of coatings under anaerobic condition was much less than that under the aerobic environment, indicating a significant delay of O_2_ reaction on the completely curable surface of the coatings. This could be probably because the system contained a large number of low degree polymerization oligomers or even non-reactive monomers because the existent oxygen consumed a part of free radicals, causing larger viscosity of the system and slower rate of the polymerization reaction.

##### ③ Gel Content

Gel content tests were applied to measure the curing and crosslinking degree of the formulations (Table 5) [23]. A great difference in the gel content existed in films cured under aerobic and anaerobic conditions, respectively. The curing rate of coatings under the aerobic condition was two times faster than that under anaerobic conditions. Besides, there was an obvious blocking effect on the polyurethane acrylate curing system with a higher degree of cure. Therefore, it was essential to eliminate the interference of O_2_ in the actual operation process to ensure the curing efficiency and performance of films.

#### 3.3.2. Mechanism of O_2_ Inhibition

Generally, a small amount of O_2_ is enriched on the surface of coatings during the painting process and some even penetrated into the coatings. The presence of O_2_ during curing process usually has a negative impact on the cured coating properties due to oxygen inhibition. The overall mechanism in free radical polymerization is shown in Figure 9. The active radicals would be formed by the decomposition of the photoinitiators under UV light, which not only further induce monomers to form monomer radical initiators, but participate in the reaction of oxygen molecules to quickly generate the peroxy radicals being partly depleted by trialkylamine. Fortunately, most active free radicals could be free from the influence of oxygen and trigger a chain reaction of the macromolecular polymers. Further, the monomer radical initiators react with oxygen molecules to form peroxy radicals, which would be completely consumed by trialkylamine and further react with O_2_ to generate peroxidation aminoalkyl radical. The entire effect was proposed to be a competition between the monomers and oxygen molecules with radicals. Consequently, strong O_2_ inhibition was expected due to the scavenging of monomers and the quenching of live radicals, leading to the decrease in the polymerization rate. The experimental results were also consistent with the proposed O_2_ inhibition mechanism.

In order to reduce the influence of O_2_ inhibition, some precautions must be taken, such as increasing the amount of initiator as well as the ultraviolet light intensity to increase the concentration of free radicals, which indicates a high reaction probability with monomers. Meanwhile, preventing oxygen access to the coating film is also worth considering. Thus, seeking a cost-effective method of deoxidization to obtain high-quality films is quite important.

#### 3.3.3. Prevention of O_2_ Inhibition

Considering the mechanism of O_2_ inhibition, the most straightforward approach is to increase the UV intensity or the amount of initiators to improve the curing conditions. This is not practical as it will easily cause the formation of a yellow production film, and accelerate the aging of the coatings when the ultraviolet light is too strong. During the polymerization, the relationship of decomposition rate *r_i_,* polymerization rate *Rp* with initiator concentration, and light intensity was respectively represented by the following formulas [24].
r_i_ = φ_i_I_0_(1-e^−2.3^^εI/[pi]^),(2)
Rp = (kp/kcp^0.5^)r_i_^0.5^[M],(3)
where φ_i_ is the photo-initiated quantum yield, I_0_ is the light intensity, ε is molar absorptivity, I is film thickness, kp and kcp are the rate constants of the chain propagation and the chain termination, respectively, and M is the initial concentration of the monomer. 

The actual measures adopted include physical and chemical methods. The physical methods, mainly from the viewpoint of effecting isolation from the contact with O_2_ and coatings, is to ensure that the UV system is cured under anaerobic conditions without the influence of the paint absorbing ultraviolet light. For example, adding a small amount of wax to the cured system to naturally produce a gel substance, which surfaced by the radiation effect in the early days, forming a protective film so as to cut off the oxygen. However, the disadvantage is that the wax density could decline when the temperature is too high, causing a protection layer that is loose for the oxygen to infiltrate arbitrarily into. The schematic of the curing process is showed in Figure 10. Another frequently used method is using nitrogen or argon, the limiting factors for it are the complex technological requirements, costly instruments and difficult operation.

The chemical methods include enhancing the odds of the reaction by radicals and monomers by changing the structure of the pre-polymer to reduce the sensitivity of the O_2_ and monomer, or by adding antioxidant ingredients to deplete O_2_ integrated into the UV-cured system such as the introduction of allyl ether or thiol in the polyester main chain to keep the modified PUA away from oxygen. Besides, inducing a strong proton donor (an active hydrogen), such as a tertiary amine, can distinctly activate oxygen free radicals to reduce the inhibition. 

In this study, we adopted the formula of UV curing coatings containing hydrogen-abstraction type photoinitiators, added with a tertiary amine as a photoinitiator aid. Here, we considered benzophenone (BP) and 4-dimethylaminophenyl-ethyl benzoate (EDAB) as an example. The mechanism of tertiary amines is shown in Figure 11. On the one hand, EDAB, as the hydrogen donor, reacted with BP and the radical initiator to produce new radicals. On the other hand, it starts a reaction with superoxide radicals or tertiary amine radicals, with the participation of O_2_, to generate peroxide aminoalkyl radicals, which may further form peroxides and new tertiary free radicals with a tertiary amine to trigger radical polymerization. Therefore, EDAB assisted the antioxidants to absorb oxygen gas.

The reason why EDAB can play a dual role in both free radical generation and oxygen consumption as the antioxidants result from the active α-H, which is mainly due to the involvement of lone pair electrons on the nitrogen atom in the hybrid of the benzene ring, coupled with the more stable tertiary radical due to stronger ability of the withdrawing group of benzoic electron.

The disadvantages of chemical methods are that the long pre-chemical duration may affect the curing efficiency, which means that it will take some time to deplete the O_2_ for the antioxidant, while the too complicated and expensive method of modifying the monomer is not conducive to large-scale industrial production. However, it was still of great significance to synthesize novel antioxidant monomer for UV curable coatings.

In this study, the process of the cover-film method, which involves covering with a layer of cling film while spraying is completed, was performed, effectively blocking the contact of oxygen with the surface of the coating without impeding incident ultraviolet light, resulting in efficiently protect UV-cured coatings against oxygen inhibition.

#### 3.3.4. Mechanism Analysis of g-C_3_N_4_

Taking the photo initiator BP/EDAB, photo induced polymerization of acrylic acid, for an example, a schematic illustration of the possible mechanism of g-C_3_N_4_ in the system containing hydrogen abstraction-type initiator composed of benzophenone (BP) and EDAB was proposed (Figure 12).

When the g-C_3_N_4_/UV coatings were irradiated by a photon of sufficient energy, equal or larger than band gap, the valence electrons (e^−^) are excited to the conduction band (CB), creating holes (h^+^) in the valance band (VB). There were sufficient basic amino groups partial oxidized oxygen-containing groups, playing the role of active sites on the surface of g-C_3_N_4_ and narrowing the effective distance due to a certain attraction in the polar functional groups of initiator and additives, by which the mechanism of photo-generated electrons and holes was supported.

Owing to the thin-layer structure (2–3 nm) of g-C_3_N_4_, the photo-generated electrons and holes could be easily produced and released from the obstructions between layers and commodiously capture the O_2_ on the surface of coatings. The O_2_ would be partly reduced to superoxide radical ion O_2_^−^, forming a conjunctive radical with BP as initiators for free radical polymerization. Besides, the photo-generated electrons can also result to the H_2_ aldehyde reductive of EDAB transferring to tertiary amine radicals, after which a binding reaction was well completed with another O_2_. Consequently, it promoted the generation of tertiary amine radicals and accelerated the O_2_ consumption at the same time. Owing to this process, an appropriate integration of g-C_3_N_4_ and UV-cured matrix would give rise to enhance the antioxidant effect of EDAB.

According to the molecular structure, the g-C_3_N_4_ was made up of sp^2^ carbon atoms, so there were lots of π bonds including π electrons, which offered plenty of space for charge carrier migration and the similar function and effect were offered by photo-generated holes. Thus, the oxidative dehydrogenation of BP would be stimulated and further reacted through hydrogen nuclei (protons) coming from the reduction of EDAB to generate a radical initiator, and which participate in the polymerization reaction.

## 4. Conclusions

The g-C_3_N_4_ was successfully incorporated into the UV-curing system after various processes including magnetic stirring and ultrasonic dispersion coupled with high shear mixing. The SEM micrograph showed that the optimal ratio of doped g-C_3_N_4_ was 0.5%–2%. The average particle size of the sample with 0.5% g-C_3_N_4_ was approximately 500 nm. The FTIR spectra also illustrated that there was a perfect amalgamation of g-C_3_N_4_ in the UV system when 0.3% g-C_3_N_4_ was added. Besides, the tests of the physicochemical and thermal properties of the coatings all showed a favorable improvement of coatings by adding g-C_3_N_4_. The pencil hardness increased to 6H. The adhesion could be 1 level and the gel content increased by 7 percent. Additionally, the decomposition of hard segments of PUA could be greatly protected by g-C_3_N_4_, leading to an enhancement of thermal stability. In addition, g-C_3_N_4_ coordinately gave rise to an enhancement in the antioxidant effect of EDAB for its promotion of the generated tertiary amine radicals and the acceleration of the oxygen consumption.

## Figures and Tables

**Figure 1 materials-13-00153-f001:**
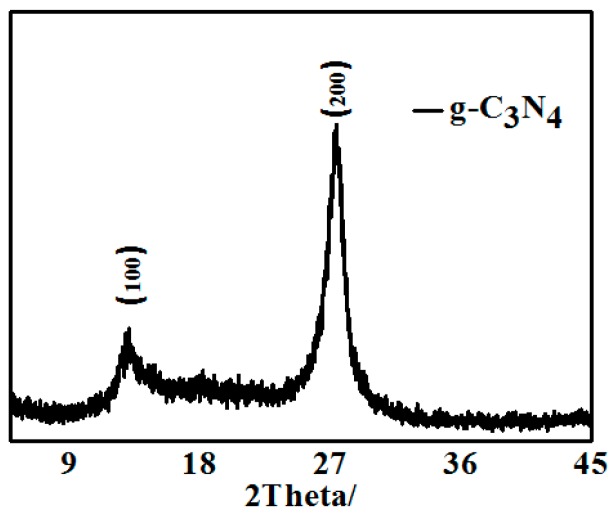
XRD pattern of pure g-C_3_N_4_ obtained by thermal decomposition.

**Figure 2 materials-13-00153-f002:**
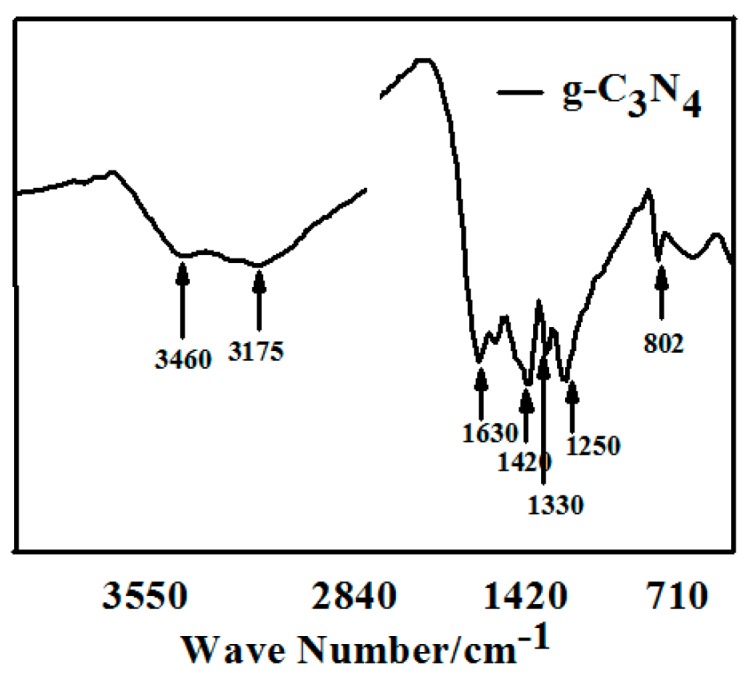
FT-IR spectrum of pure g-C_3_N_4_ prepared by semi-closed pyrolysis.

**Figure 3 materials-13-00153-f003:**
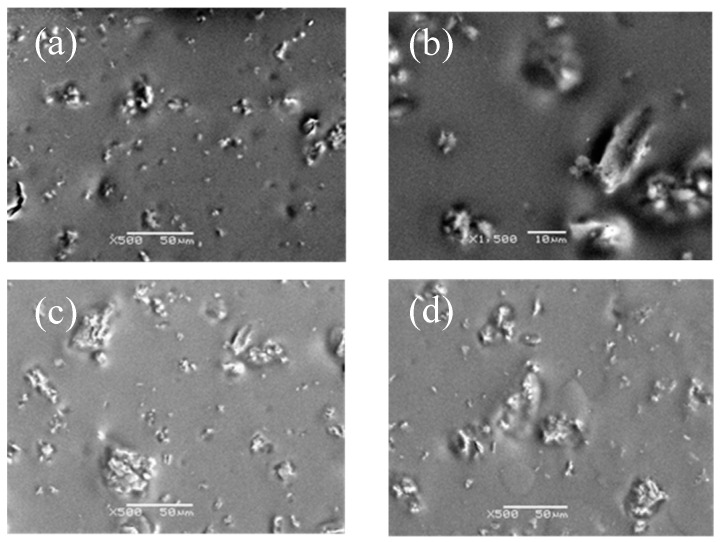
SEM micrograph of compound coatings with various content of g-C_3_N_4_: (**a**) 0.5% g-C_3_N_4_; (**b**) 1% g-C_3_N_4_; (**c**) 1.5% g-C_3_N_4_; and (**d**) 2% g-C_3_N_4._

**Figure 4 materials-13-00153-f004:**
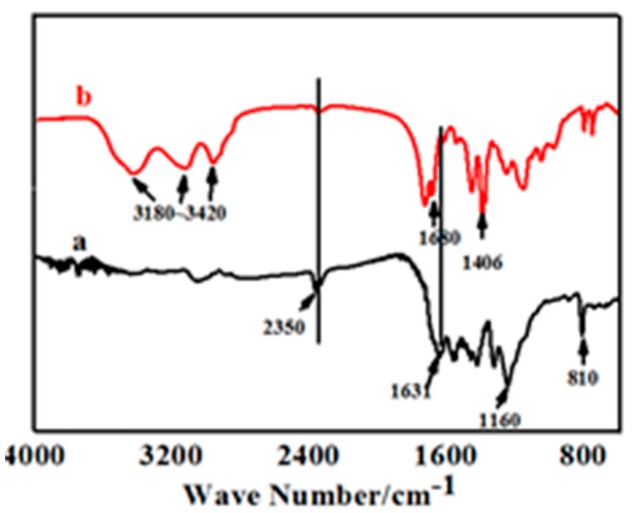
FT-IR spectrum of UV-curing coatings added with 0.3% of g-C_3_N_4_ (**a**) and none (**b**).

**Figure 5 materials-13-00153-f005:**
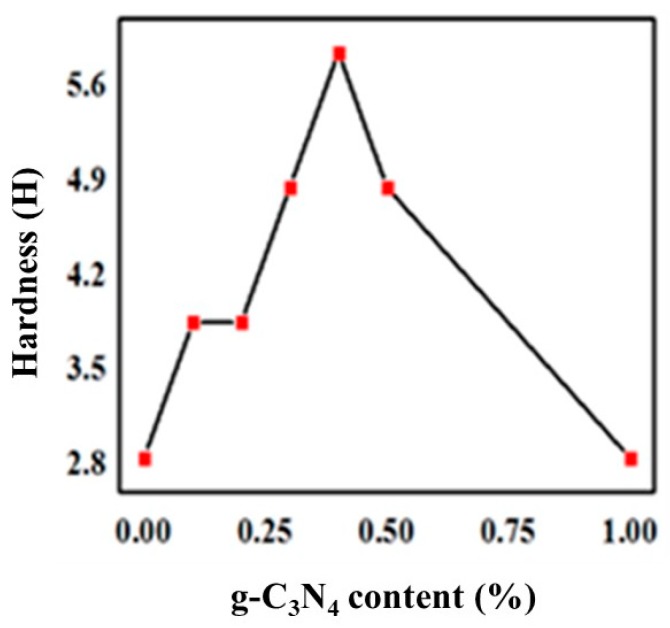
Variation of hardness with g-C_3_N_4_ content.

**Figure 6 materials-13-00153-f006:**
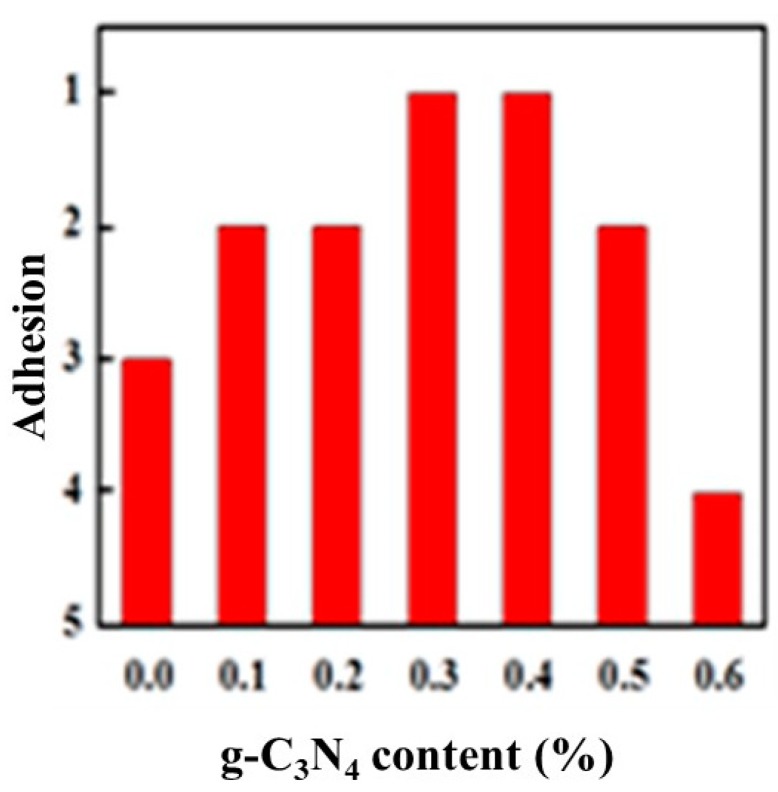
Variation of adhesion with g-C3N4 content.

**Figure 7 materials-13-00153-f007:**
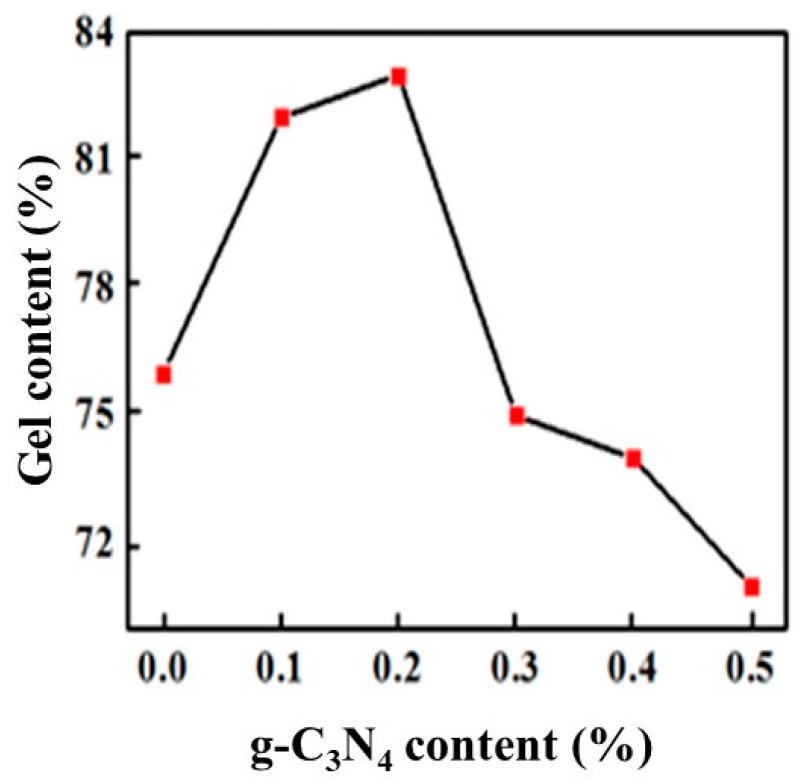
Variation of gel content with the proportion of g-C_3_N_4._

**Figure 8 materials-13-00153-f008:**
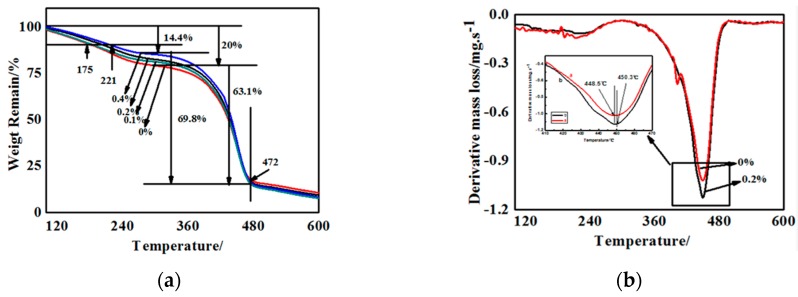
Thermo gravimetric (TG) and differential thermal gravity (DTG) curves of g-C_3_N_4_/PUA compound coatings containing various content of g-C_3_N_4_. (**a**) Thermo gravimetric; (**b**) Dfferential thermal gravity.

**Figure 9 materials-13-00153-f009:**
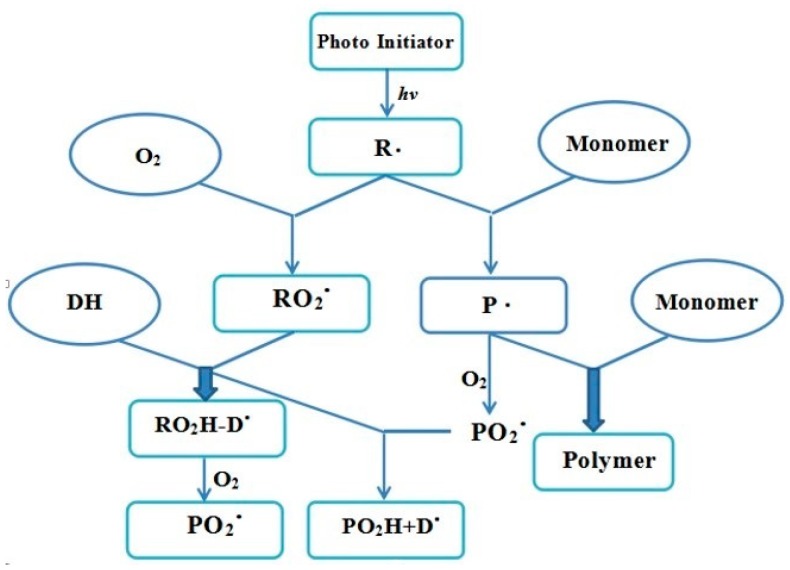
Mechanism of O_2_ inhibition in free radical polymerization. Note: R: living radical; P: monomer radical initiator; PO_2_: monomer peroxy radicals; RO_2_: initiator peroxy radicals; DH: trialkylamine; PO_2_H: peroxidation monomer; DO_2_: peroxidation aminoalkyl radical.

**Figure 10 materials-13-00153-f010:**
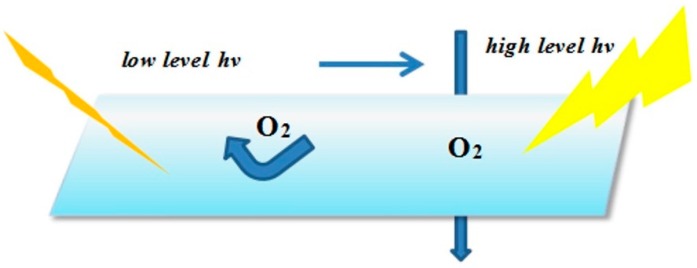
Schematic of physical methods of isolating from oxygen.

**Figure 11 materials-13-00153-f011:**
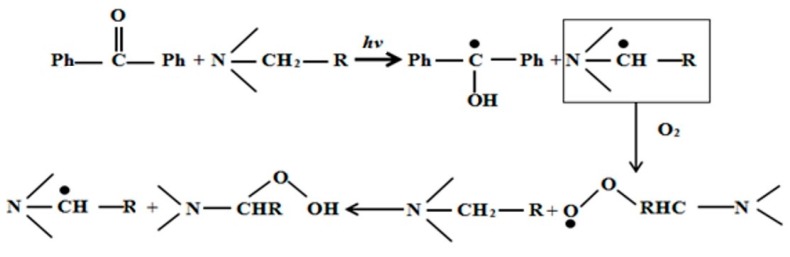
Mechanism of decreased O_2_ inhibition.

**Figure 12 materials-13-00153-f012:**
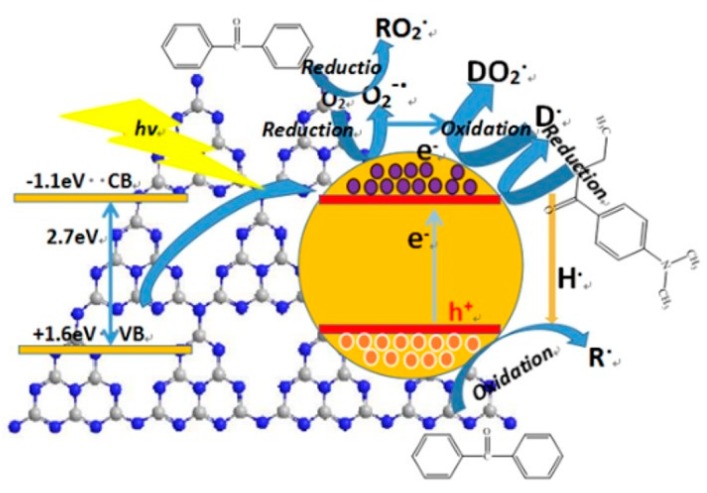
The mechanism of g-C_3_N_4_ on the system containing hydrogen abstraction-type initiator.

**Table 1 materials-13-00153-t001:** Acid resistance of g-C_3_N_4_/UV compound system.

Time/h	2	4	6	8	10
None g-C_3_N_4_	unchanged	unchanged	soften, little exfoliated	badly worn	entirely exfoliated
0.3% g-C_3_N_4_	unchanged	unchanged	soften	little worn	badly worn

**Table 2 materials-13-00153-t002:** Alkali resistance of g-C_3_N_4_/UV compound system.

Time/h	2	4	6	8	10
None g-C_3_N_4_	unchanged	unchanged	slightly softened	soften, little exfoliated	badly worn
0.3% g-C_3_N_4_	unchanged	unchanged	unchanged	unchanged	soften, unexfoliated

**Table 3 materials-13-00153-t003:** Hardness and adhesion of the coatings with different ratios cured under aerobic and anaerobic conditions.

	Pure Coating (Aerobic)	Pure Coating (Anaerobic)	Composite Coating (Aerobic)	Composite Coating (Anaerobic)
Pencil hardness/H	2	3	5	6
Adhesion/class	3	3	1	2

**Table 4 materials-13-00153-t004:** Effect of O_2_ on tack free time.

Coatings	Pure Coating (Aerobic)	Pure Coating (Anaerobic)	Composite Coating (Aerobic)	Composite Coating (Anaerobic)
Tack Free Time/s	680	260	730	270

**Table 5 materials-13-00153-t005:** Effect of O_2_ on gel content.

Coatings	Pure Coating (Aerobic)	Pure Coating (Anaerobic)	Composite Coating (Aerobic)	Composite Coating (Anaerobic)
Gel content/s	40	78	41	73

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
