# Peer review of "Effect of Graphene Carbon Nitride on Ultraviolet-Curing Coatings"

_materials, 2019, doi:10.3390/ma13010153_

Round 1

Reviewer 1 Report

Dear Authors,

I have read your manuscript carefully and I would say that this manuscript would be very interesting for readers. The objectives of the study are clearly defined. The introduction provides a good, generalized background of the topic. The results are clearly explained and are presented in an appropriate format. The figures show essential data; some of the data are also summarized in the text. I do not think any additional graphics are necessary. The cited literature is relevant to the study and balanced. This manuscript was well prepared and could be published after correcting some editing errors.

The reviewer recommends using “SEM micrograph”, not “SEM image” in line 163 and 458 and figure 3,

In line 31 a letter “a” should be removed.

In line 76 the dash from word “oxy -gen” should be removed.

In line 203 space between “g-C3N4” and “means” is needed.

Author Response

Response to Reviewer 1 Comments

Thank you very much for your valuable and careful comments. We have made corrections carefully according to your comments. I will answer your suggestions and questions one by one below. Special thanks to you for your good comments.

Points: The reviewer recommends using “SEM micrograph”, not “SEM image” in line 163 and 458 and figure 3.

In line 31 a letter “a” should be removed.

In line 76 the dash from word “oxy -gen” should be removed.

In line 203 space between “g-C3N4” and “means” is needed. 

Response: Thank you for your careful comments. The changes and modifications of the manuscript have been highlighted in red through the text of the revised manuscript.

Reviewer 2 Report

The authors successfully demonstrated the process for graphene carbon nitride incorporated into the ultraviolet (UV)-curing system. They provided detail characterization of graphene carbon nitride. They systemically elucidate hardness, stability and degree of curing depending on gel contents. if only the specific data are reinforced, it is expected that it will have a good effect in this field by suggesting a applicable method of making the stable graphene carbon nitride films. Hence, I recommend that this manuscript is enough to publish this journal “Materials” after major revision as follows;

English used in the manuscript at many places is poor. The author should improve the English. In Figure 5, 6 and 7, authors should provide x-axis and y axis titles in each plot and clearly define the meaning in the manuscript. The author mainly describes various data related to g-C3N4. 0.3%. But The author should give SEM images of g-C3N4. 0.3%. in Figure 3 In Figure 5, 6 and 7, error bar should be inserted in each plot.

Author Response

Response to Reviewer 2 Comments

Thank you very much for your valuable and careful comments. We have made corrections carefully according to your comments. I will answer your suggestions and questions one by one below. Special thanks to you for your good comments.

Points: English used in the manuscript at many places is poor. The author should improve the English. In Figure 5, 6 and 7, authors should provide x-axis and y axis titles in each plot and clearly define the meaning in the manuscript. The author mainly describes various data related to g-C3N4. 0.3%. But The author should give SEM images of g-C3N4. 0.3%. in Figure 3 In Figure 5, 6 and 7, error bar should be inserted in each plot. 

Response: Thank you for your careful comments. The English in the revised manuscript has been edited and improved by a native English speaker. The places that need to be modified have also been modified in the article. Data such as the infrared spectrum of 0.3% g-C3N4 are given in the article to illustrate related issues.

Round 2

Reviewer 2 Report

I am satisfied with the revision